



# Development and use of a lightweight sampling system for height-selective drone-based measurements of organic aerosol particles

Christine Borchers[1], Lasse Moormann[2], Bastien Geil[1], Niklas Karbach[1] and Thorsten Hoffmann[1]

[1]Department of Chemistry, Johannes Gutenberg University, Mainz, Germany

[2]Multiphase Chemistry Department, Max Planck Institute for Chemistry, Mainz, Germany

*Correspondence to*: Thorsten Hoffmann (t.hoffmann@uni-mainz.de)

## Abstract

Organic aerosols (OA) are introduced into the atmosphere from a variety of natural or anthropogenic sources. Especially in the sub micrometer range, the organic fraction contributes to a large proportion of the particle mass and thus has an impact on

climate and air quality. To gain insights into sources and sinks and the significance of dispersion, mixing and ageing processes for OA, vertical profiling of the concentration of organic aerosols is particularly helpful. Therefore, the aim of this study is to present an aerosol particle sampler that is suitable to be used onboard uncrewed aerial vehicles (UAVs). The sampler consists of a three-dimensionally printed filter holder connected to a lightweight high-performance pump that can generate a flow rate of up to 19 L min$^{-1}$ for up to 30 minutes. The sampler was characterized and applied on a proof-of-concept study during the

BISTUM23 campaign in August 2023 in Southern Germany. Vertical profiles were measured with three samplers mounted on ground and drones and collected aerosol particles in an altitude of 1.5 m, 120 m and 500 m above ground level simultaneously. The filters were analyzed with UHPLC-HRMS, and a targeted approach was used to determine vertical profiles and diurnal trends of biogenic, anthropogenic and biomass burning marker compounds. A non-targeted analysis revealed a high number of CHO-containing compounds, which were oxidized to a greater extent during the course of the day and at increasing altitudes.

The system presented here provides a comparatively simple and cost-effective way to sample OA at different altitudes and at different locations and thus obtain vertical concentration profiles of the organic aerosol composition.

## 1 Introduction

Organic aerosol particles (OA) are accountable for a large proportion (20–90%) of the sub-micron particle mass in the lower troposphere. They affect air quality, climate and human health (Benoit et al., 2023; Jimenez et al., 2009; Kanakidou et al.,

2005). Primary organic aerosols (POA) are emitted directly, for example from biogenic sources such as plant debris or in the form of spores, bacteria or viruses, or from sources that are mostly anthropogenic such as combustion processes. Secondary organic aerosols (SOA) are formed by oxidation of volatile organic precursors and subsequent condensation of the products (gas-to-particle conversion) (Reddington et al., 2011; Kroll and Seinfeld, 2008; Kerminen et al., 2005). The chemical composition of OA provides information on the individual sources and source processes (Gouw and Jimenez, 2009).





Nitroaromatic compounds, for example, are released during the combustion of coal or wood, and are contained in vehicle exhaust gases. They can also be formed as secondary products from the reaction of phenols or cresols with $NO_x$ (Wang et al., 2020; Harrison et al., 2005; Lu et al., 2019). The combustion of lignocellulose, the most abundant biomass resource on Earth, leads to the production of phenolic compounds with aldehyde functionalities, including 4-hydroxybenzaldehyde, vanillin and syringaldehyde. Phenol aldehydes can be oxidized in the atmosphere by OH radicals, $NO_3$ radicals or ozone, leading to the

production of carboxylic acids (Cao et al., 2022; Rana and Guzman, 2022; Net et al., 2011). Vegetation on Earth emits large amounts of volatile organic compounds (VOCs) such as isoprene and various monoterpenes (MTs), with the most important MT, α-pinene, accounting for about one third of global MT emissions (Sindelarova et al., 2014). In the atmosphere, oxidation by OH radicals, $NO_3$ radicals or ozone leads to various products that differ in their volatility by several orders of magnitude. Products such as 2-methyl tetrols, terpenylic acid, terebic acid, and pinonic acid are described as the main oxidation products

(Kołodziejczyk et al., 2020; Bianchi et al., 2019; Nozière et al., 2015; Müller et al., 2012; Kroll and Seinfeld, 2008; Claeys et al., 2004; Hoffmann et al., 1997). It can be concluded that the elucidation of the chemical composition of OA can provide valuable information about the sources, source strengths and processing of organic aerosol components, such as the contribution of biogenic sources in terrestrial ecosystems or the role of anthropogenic contributions to organic aerosols.

The implementation of atmospheric concentration measurements in the form of vertical gradient measurements offers several advantages: firstly, measurements at multiple heights allow the identification of sources and sinks, as they can distinguish between local plumes and emission sources at ground level and atmospheric background concentrations. The latter are particularly characterized by aging in the case of OA (Li et al., 2024). Vertical profiling can also be used to investigate the transport and distribution of OA. Therefore, many monitoring stations also perform atmosphere-related observations from tall

towers, as they enable measurements at several heights within the planetary boundary layer or even beyond, and can thus reflect both local processes at lower altitudes and regional influences at higher altitudes (Li et al., 2024; Mikhailov et al., 2017; Andreae et al., 2015; Williams et al., 2011). This is where drones can also be used without the need for a suitable tower infrastructure. The use of miniaturized sampling systems in conjunction with drones has attracted considerable attention in recent years (Böhmländer et al., 2024). Compared to conventional sampling methods that use towers, balloons or aircraft, these

systems offer the advantages of smaller size, environmental friendliness and the ability to collect samples in remote locations that are difficult to access (Thivet et al., 2024; Pusfitasari et al., 2022; Lan et al., 2020; Bieber et al., 2020).

The aim of this proof-of-concept study was therefore to develop a new low-cost and lightweight aerosol sampling system for drones. To do this, a filter holder was designed and produced using 3D printing and connected to a lightweight high-

performance pump. The sampled filters were then extracted and analyzed using UHPLC-MS. The system was characterized and vertical concentration profiles of OA compounds at different times of the day were determined as part of the BISTUM23 campaign in August in the Swabian Jura, Southern Germany.





## 2 Experimental procedures

### 2.1 Sampling System

For aerosol sampling, a home-made 3D-printed filter holder was connected to an electric fan motor with an impeller (CDS-R540-QA012; DC 7.2 V; 70 W; SIP Cinderson Motor CO., LTD), which enables high gas flow rates. The inlet diameter of the filter holder was 20 mm. The filters with a diameter of 70 mm were placed on a stainless steel mesh to prevent the filter from tearing even at high flow rates (see Figure 1). The sampling unit was powered by a lithium polymer battery (LiPO 7.4 V; 5000 mAh; Conrad Energy), which allows an operating time of about 30 minutes, which is slightly above the approximate

maximum flight time of the drones on which the sampler is mounted on. The total weight of the filter holder and electric blade motor is 280 grams, which corresponds to the weight of the battery.

To check the stability of the flow rate through the sampling unit, a filter holder (equipped with PallflexTM EmfabTM filters TX40HI20WW, 70 mm) was connected to a flow meter (model 4148, TSI GmbH, USA) and the flow rate was recorded at 30-

second intervals over a total period of 30 minutes.

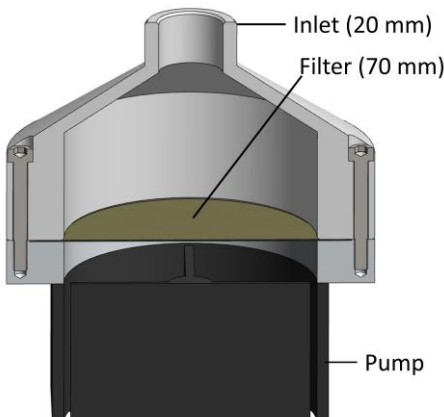

**Figure 1: Schematic representation of the filter holder. The air containing aerosol particles is sucked through the filter, which is**
**positioned on a stainless steel mesh.**

### 2.2 Sampling procedure

The filter holders were attached below the drones with the filter opening facing to the side (Fig. S1). During a deployment near the village of Essenheim near Mainz (49° 55' N, 8° 10' E), the drones (model Matrice 200 and Matrice 300, both DJI) and the

third filter holder were operated at the same height, approx. 5 m above the ground, and with a horizontal distance of approx. 5 m. Borosilicate glass microfibers, reinforced with glass cloth and bonded with PTFE (PallflexTM EmfabTM Filters



TX40HI20WW, 70 mm), were used for aerosol sampling. Sampling time was between 20 and 30 minutes depending on the battery power of the drones. To conduct the vertical profile measurements, the system was operated near the village of Albstadt, in the Swabian Jura (48° 15' N, 8° 59' N). The two drones were simultaneously operated at 120 m and 500 m (one above the

other with a horizontal offset of about 10 m), while the third filter holder was attached to a wooden frame at a height of about 1.5 m above the ground, with again a vertical offset of about 10 m (Fig.S2). A third measurement drone, FLab (Moormann et al., 2024), was used to simultaneously quantify gas tracers and meteorological data in hourly vertical profiles over the course of a day at the same measurement location.

**2.3 Analysis**

The filters were stored at -25 °C until analysis. The filters were extracted according to the following protocol. They were cut into small pieces and placed in a vial that had been previously baked at 450 °C for at least 8 hours. They were then extracted with 3 mL and twice 1.5 mL of a 9:1 (v/v) mixture of LC/MS grade methanol (Carl Roth) and LC/MS grade water (Thermo Fisher Scientific) for 30 minutes on a shaking plate. The supernatant was transferred to a 1.5 mL HPLC vial and concentrated at 30 °C under a gentle $N_2$ stream. The residue was then filtered with a PTFE filter (pore size: 0.20 µm; Altmann Analytik)

and filled up to about 50 µL with water. Since the volume of the solution is not known, a camphor sulfonic acid standard was added to obtain a correction factor for potential volume discrepancies (See Supplement S1).

Analysis was performed in triplicate using a Dionex UltiMate 3000 ultra-high-performance liquid chromatography system coupled to a heated electrospray ionization source (HESI) and a high-resolution Q-Exactive Orbitrap mass spectrometer (HRMS) (all Thermo Fisher Scientific). An Acquity UPLC CSH Fluoro Phenyl (PFP) column, 100 mm×2.1 mm with 1.7 µm

particle size (Waters) was used for chromatography. The eluent A was 98% LC/MS grade water (Thermo Fisher Scientific) with 0.04% formic acid and acetonitrile (VWR Chemicals), the eluent B was 98% acetonitrile and water, and the injection volume was 5 µL. An $H_2O$/ACN gradient was used for the analysis. A flow rate of 0.5 mL min$^{-1}$ and a gradient as described below was used: Starting with 10% B, increasing to 99% B in 11 min, after which B was held at 99% for 1 min, decreased to 10% in 0.5 min, and held again for 0.5 min. The HESI source was used in negative mode, resulting in the formation of

deprotonated molecular ions. The sheath gas and auxiliary gas pressures were 40 and 20 a. u. (arbitrary unit) respectively. The auxiliary gas heater temperature was 150 °C and the capillary temperature was 350 °C. The sprayer voltage was set to -4.00 kV. Further details on the additional chemicals used, including their respective purities, can be found in Supplement S2.

**3. Results and discussion**

**3.1 Sampler characteristics**

Figure 2 shows the airflow through the filter holder as a function of time. During the measurement, the recorded airflow decreases from 19.0 L min$^{-1}$ to 17.8 L min$^{-1}$. This may be due to the fact that no voltage regulator was installed between the battery and the motor, so the voltage in the battery decreases over time, and thus the power of the motor also decreases. A



statistically significant (significance level $\alpha = 0.05$) linear relationship was obtained between flow rate and time. A linear fit ($y = mx + b$) was performed, with $m = (-0.027 \pm 0.002)$ L min$^{-2}$ and $b = (18.97 \pm 0.04)$ L min$^{-1}$ as fit parameters. This function can then be used to determine the volume of air collected within the sampling time (see Supplement: Determination of the concentration).

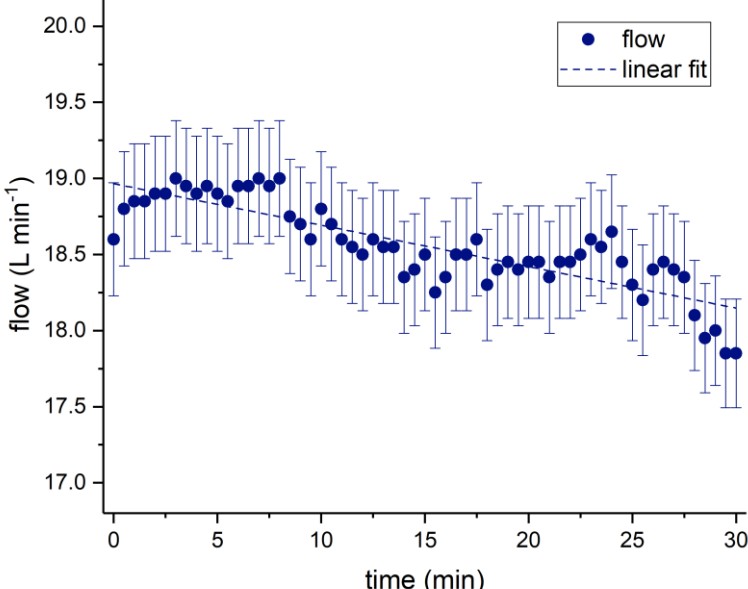

**Figure 2: Flow through the filter holder as a function of time (blue dots), the dashed line represents a linear fit of the data. The errors correspond to 2% of the measured value, which represents the measurement uncertainty of the flow meter.**

Most aerosol particle collectors operate at flow rates between 10 and 500 L min$^{-1}$ to collect aerosol particles over periods of several hours or days (Ma et al., 2022; Leppla et al., 2023). Since the presented system is designed for use onboard drones, a lightweight configuration is crucial. These constraints result in an operational time of 20 to 30 minutes and a flow rate of approximately 18 L min$^{-1}$. However, sample preparation is essential to be able to detect individual components despite these restrictions. Due to the extraction method and the reduction of the sample volume to only 50 µL of liquid, it is possible to detect a wide range of biogenic and anthropogenic substances. Figure 3 shows an excerpt of an extracted ion chromatogram (EIC) of a LC-MS run for *m/z* 185.0819 (red line) and *m/z* 157.0506 (blue line), which originate from a loaded and a blank filter (red and blue dashed lines). The mass traces refer to biogenic maker substances. As the mass spectrometer was operated with a HESI ion source in negative mode, the EICs demonstrate the deprotonated compounds. The signal observed at a retention time of approximately 2 min in the EIC at *m/z* 157.0506 can be attributed to terebic acid, which is an oxidation product of α-pinene (Kołodziejczyk et al., 2020). The second mass trace (*m/z* 185.0819) shows several signals. The occurrence of these signals can be attributed to several constitutional isomers of pinic acid, an oxidation product of α-pinene, which exhibit identical mass-to-charge ratios. However, not only α-pinene is emitted in the atmosphere, but also various other terpenes such as 3-carene, sabinene or limonene. These terpenes oxidize and form compounds such as 3-caric acid, sabinic acid, or limonic





acid, which have the same sum formula as pinic acid. This results in the occurrence of different compounds at a single mass

trace (Glasius et al., 2000).

It is evident that the signals of the loaded filters differ by an order of magnitude from those of the blank filter and that a high signal-to-noise ratio is achieved for both mass traces. This demonstrates that with the light aerosol sampling system presented here, in conjunction with the extraction method described, it is possible to detect and quantify various marker substances despite the relatively short sampling time. As a result, the presented system is ideally suited for use onboard drones.

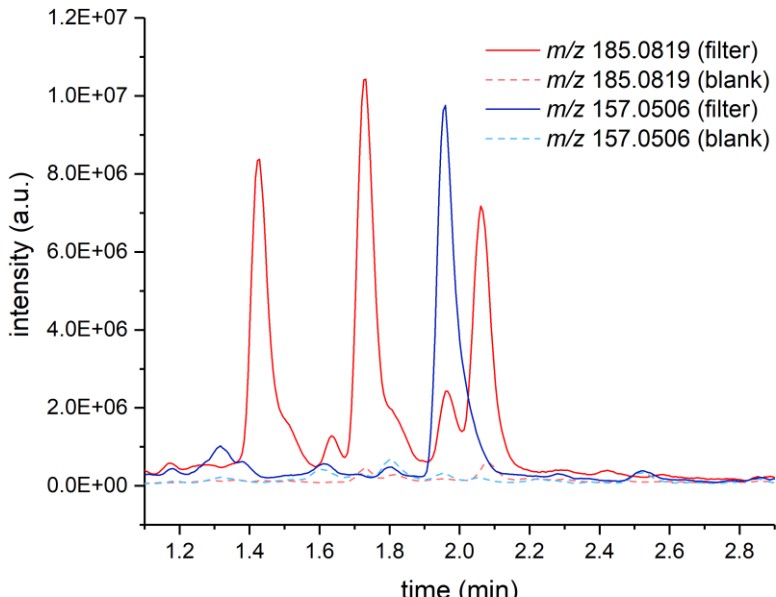


**Figure 3: Extracted ion chromatogram for m/z 185.0819 (red line) and m/z 157.0506 (blue line) for a loaded filter and the corresponding EIC for a blank filter (red and blue dashed line). This EIC is representative of biogenic marker compounds like pinic acid, limonic acid, sabinic acid, 3-caric acid (red line), or terebic acid (blue line).**

### 3.2 Influence of the sampling drone on the measured concentrations

To evaluate whether differences in the drone models or sample device mounting positions (see Fig. S1) impact analysis results (e.g., due to aspiration flow variations), two sampler onboard drones and one at a metal framework were operated simultaneously at ~5 m height near Essenheim during a test flight. Figure 4 compares the concentrations of a few selected exemplary compounds (pinic acid, 4-nitrophenol, terebic acid and 2,6-dimethyl-4-nitrophenol) at different sampling times for the respective systems. The results of these replicate measurements at the three different sampling periods are shown in brown,

blue and green. According to these results, the concentrations of the individual compounds fluctuate noticeably between the different sampling periods, which is plausible due to the approximately one-hour delay between the three sampling periods. The differences between these measurement flights can be attributed to the different time of the flights and the resulting



changes in the composition of the collected aerosol particles. The discrepancy between the second and third measurements is
particularly evident for the anthropogenic markers 4-nitrophenol and 2,6-dimethyl-4-nitrophenol. Such fluctuations can be
attributed, for example, to a change in wind direction and thus to a different origin of the sampled air mass. However, it is
crucial that the differences between the different sampling systems within a measurement flight are comparatively low for the
analytes. It can be concluded that the type of drone used, or minor differences in the mounting position of the sampler on the
drone and the resulting differences in air turbulence around the filter holder, have no significant influence on the analytical
results.


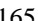

**Figure 4: Mean concentration of pinic acid, 4-nitrophenol, terebic acid and 2,6-dimethyl-4-nitrophenol, for the three measurement
setups (no drone; dji m300; dji m200) during three measurement flights (brown, blue and green). The error bar is the result of two
error sources: the standard deviation derived from the triple determination made by the LC-MS measurement; and the error
associated with the flow measurement of the filter holder.**



**3.3 Vertical profiles of biogenic, biomass burning and anthropogenic marker compounds**

The characterized filter sampler was used to sample aerosol particles simultaneously to measurements of a third measurement drone, FLab (Moormann et al., 2024) during the BISTUM23 campaign in August 2023 in Albstadt, Germany. This approach allowed for the acquisition of daily and height trends for OA at a measurement site, which is surrounded in all directions by a mixture of grassland, forest, agricultural land, and urban infrastructure (see Fig S2).

Figure 5 shows three height profiles at 10:35 am, 1:35 pm and 4:30 pm local time (UCT+2) for concentrations of the three biogenic marker components pinic acid (black square), terpenylic acid (blue triangle) and terebic acid (green diamond) in 1.5 m, 120 m and 500 m.

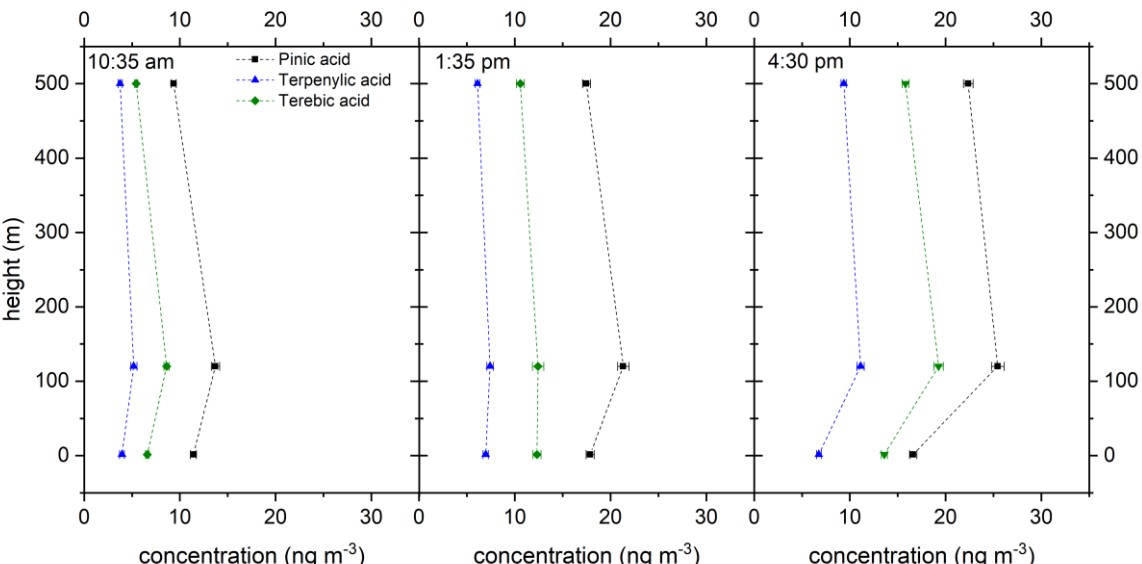

**Figure 5: Vertical profiles of the biogenic marker compounds pinic acid (black square), terpenylic acid (blue triangle) and terebic acid (green diamond) at the different times (all times are in UTC+2). For better clarity, these points are connected by dashed lines The error bar is the result of two error sources: the standard deviation derived from the triple determination made by the LC-MS measurement; and the error associated with the flow measurement of the filter holder.**

It can be seen that the concentrations at a height of 120 m are higher than at ground level (1.5 m). This observation can have multiple causes, for example the different footprint areas attributable to the various heights or also dry deposition of corresponding aerosol-borne components on the ground (Spielmann et al., 2017; Bamberger et al., 2011). Actually, this trend between 1.5 m and 120 m also coincides with the measured ozone concentration (Fig. S3). The ozone concentration is slightly lower at ground level than in the higher region. However, the concentrations of the selected oxidation products decrease up to a height of 500 m. This finding could be attributed to the higher average residence time of the corresponding aerosol populations. One potential explanation for the concentration decrease is, among others, the further oxidation of the compounds measured here, which could lead to the formation of more highly oxidized compounds. In addition to the altitude trend, a distinct daytime trend can also be seen. From morning to afternoon, the concentrations of all three compounds increase at all



altitude levels, although the relative increase depends on the individual compound. This is also in good agreement with the measured ozone concentration. The ozone concentration increases during the day at all altitudes.

Figure 6 shows the altitude profile of some marker compounds for anthropogenic sources of OA and biomass burning, salicylic acid (purple circle), 4-hydroxybenzaldehyde (pink hexagon), 4-nitrophenol (turquoise pentagon), 2,6-dimethyl-4-nitrophenol (light blue half-filled circle), and 2,4-dinitrophenol (brown star). The actually measured concentrations are shown as symbols. For these marker substances, no clear trend can be seen in terms of altitude or time of day. It can be observed that the trend in terms of altitude or time of day is comparable for salicylic acid and 4-hydroxybenzaldehyde, as well as for 4-nitrophenol and

2,4-dinitrophenol. The differences between the trends are probably due to different sources of the marker substances. For example, salicylic acid and 4-hydroxybenzaldehyde are formed during the combustion of lignin (Cao et al., 2022; Rana and Guzman, 2022; Fleming et al., 2020). In addition, salicylic acid has been detected in vehicle exhausts, making it both an anthropogenic and a biomass-burning marker compound (Li et al., 2020). The nitroaromatic compounds may originate from the combustion of biomass and the nitration of phenols or vehicle exhaust gases (Zhang et al., 2022; Kulakova et al., 2020; Lu

et al., 2019). Consequently, they can also be considered as marker substances for biomass burning and anthropogenic substances. The highest concentrations of salicylic acid, 4-nitrophenol, 2,6-dimethyl-4-nitrophenol and 2,4-dinitrophenol were observed in the morning at a height of 120 m. This indicates that the air mass sampled at 10:35 am had crossed an area where biomass had been burned or where there was heavy traffic. The wind direction determined by FLab is southwest with a wind speed of 2 to 3 m s$^{-1}$ (Figs. S4 and S5). A federal highway is also located in this direction. The backward trajectory for the

120 m sample created with the NOAA HYSPLIT model (Rolph et al., 2017; Stein et al., 2015; Draxler and Hess, 1998) indicates that the sampled air masses crossed the federal highway at around 8 am (Fig. S6). Higher concentrations of anthropogenic markers can be attributed to rush-hour traffic. At ground level, the plume is less extended due to the surface layer (Stull, 2017), while at 500 m, it is diluted by dynamic air mixing or different origin of the air masses (Fig. S6). Thus, compound concentrations from the federal road, especially in the morning, are lower at 1.5 m and 500 m.

The measured concentrations of the anthropogenic markers are lower than those of the biogenic ones. The lower concentrations can be explained by the fact that the sampling site was in a rural area where anthropogenic influences are possibly less significant than biogenic ones.



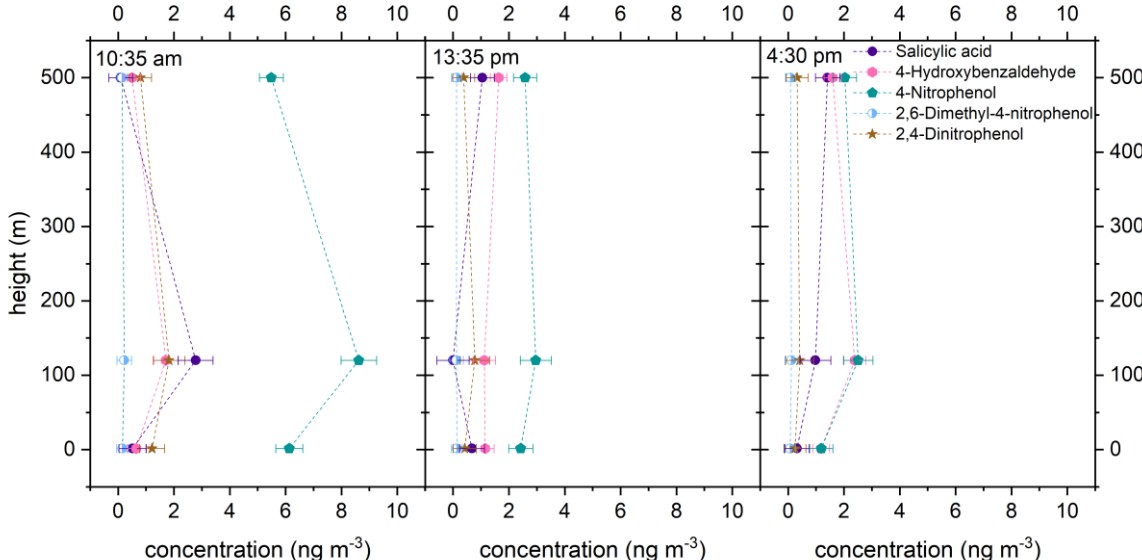

**Figure 6: Vertical profiles of the anthropogenic marker compounds salicylic acid (purple circle), 4-hydroxybenzaldehyde (pink hexagon), 4-nitrophenol (turquoise pentagon), 2,6-dimethyl-4-nitrophenol (light blue half-filled circle), and 2,4-dinitrophenol (brown star). The actually measured concentrations are shown as symbols. For better clarity, these are connected with dashed lines. The error bar is the result of two error sources: the standard deviation derived from the triple determination made by the LC-MS measurement; and the error associated with flow measurement of the filter holder.**

### 3.4 Height dependent Van Krevelen diagrams

In addition to the targeted analysis of individual marker compounds described above, a non-target analysis was also carried out. The results are shown in a series of Van Krevelen diagrams in Figure 7. The underlying molecular formulas of the compounds shown can be unambiguously assigned due to the use of a high-resolution mass spectrometer with accurate mass determination and were determined using MZmine 2.53 software (Pluskal et al., 2010). These were used to determine the H/C and O/C ratios, which were then plotted against each other. The compounds were assigned to the four substance classes CHO

(blue), CHON (green), CHOS (orange) and CHONS (pink), with CHO being the most abundant class. The size of the dots is defined by the measured peak intensity of the respective LC-MS measurement. All signals were normalized to the duration of the sampling. Since a HESI source was used as the ion source, it is important to consider the potentially different ionization efficiencies of the compounds. The ionization efficiency can differ by several orders of magnitude for differently functionalized compounds (Liigand et al., 2021; Oss et al., 2010). Therefore, the dot size essentially provides an overview of the concentration

changes of the respective compounds as a function of time or height. However, it does not provide any information about the relative amounts between different compounds. The Van Krevelen diagrams are divided into five sections for better clarity. These are based on the maximum carbonyl ratio (MCR) of the compounds. This describes the maximum contribution of the carbonyl/epoxy functionalities of the components and thus provides an indication of their degree of oxidation. The five groups are *V*: highly unsaturated (combustion related), *IV*: oxidized unsaturated (primary organic carbon, oxidation products from



aromatic VOC), *III*: intermediately oxidized (monoterpene first generation oxidation products), *II*: highly oxidized (monoterpene oxidation products, oxidative aging), and *I:* very highly oxidized (isoprene oxidation products) (Zhang et al., 2021).

The composition of the aerosols should not differ significantly regardless of the origin of the air mass due to the remote location. At first glance, however, it can be seen that the Van Krevelen diagram at a height of 120 m in the morning (Figure 7d) differs significantly from all the others. This sample shows a strikingly high number of CHO-containing compounds with high peak areas in the region between areas *III* and *IV* and in area *V*, likely originating from combustion processes. Studies link these regions of the Van Krevelen diagram to biomass combustion (Tang et al., 2020). Compounds in this area ($C_{20}H_{26}O_3$, $C_{20}H_{28}O_2$, $C_{20}H_{28}O_3$, $C_{20}H_{30}O_2$, $C_{20}H_{30}O_4$) are identified as biomass combustion markers in previous research (Ramteke et al., 2024; Smith et al., 2009). The intense biomass burning and anthropogenic tracers in the 120 m morning sample (section 3.3) may result from vehicle exhaust and biomass burning, as discussed for the vertical profiles of anthropogenic substances in Figure. 6 and Section 3.3. However, the change in the surface layer and atmospheric boundary layer heights in the morning hours is likely related to these observations. To examine general trends over height and time of day, the Van Krevelen diagram for a height of 120 m in the morning is not considered in the discussion below.

The vertical profiles at 1:35 pm and 4:30 pm are very similar. At a height of 1.5 m, there are several substances in area *IV* that are no longer present at heights of 120 m and 500 m, or are present in significantly lower concentrations. This can be attributed to the oxidation of the substances to more highly oxidized compounds. In addition, a shift of the points to higher O/C and H/C ratios is observed between 120 m and 500 m, which leads to an increased number of substances in section *II*. This also indicates that the observed compounds at higher altitudes are evolving into more highly oxidized substances. The observed tendency towards higher concentrations of higher-oxidized compounds at higher altitudes can be explained by the longer residence time in the atmosphere before reaching these altitudes. Consequently, terpenes, isoprene and their oxidation products are exposed to oxidizing species for a longer period of time, resulting in the formation of more highly oxidized compounds.

For the variation during the day, the diagrams at the same altitudes can be compared with each other. It is noticeable that the size of the points and thus the associated measured concentration increases during the course of the day, particularly for the CHO-containing compounds. This finding is also consistent with the targeted approach, which detects an increase in the concentrations of biogenic SOA compounds (in this case CHO compounds) during the course of the day. The higher concentration of CHO-containing compounds can be attributed to the rising temperature and the accumulating oxidation product concentrations over the course of the day. As the temperature rises, more biogenic substances such as terpenes are emitted by the trees, which are then oxidized to CHO-containing compounds over the course of the day (Vettikkat et al., 2023; Niinemets et al., 2004).





**Figure 7: Van Krevelen diagrams in the morning (left), at noon (middle) and in the afternoon (right). The different substance classes are represented by different colors (CHO blue; CHON green; CHOS orange; CHONS pink) The size of the dots correlate with the measured peak area of the compounds. The Van Krevelen diagrams are divided into 5 areas depending on the MCR, which are separated by dashed lines (Zhang et al., 2021).**






## 4. Conclusions

This study demonstrates that drones can also be used as a sampling platform for detailed chemical characterization of organic aerosol components using UHPLC-MS. The developed aerosol sampling unit allows the collection of sufficient aerosol mass for both targeted and non-targeted analysis of primary and secondary organic aerosol components within the maximum flight
time of the drones used. The newly developed, very light aerosol sampling system with a weight of approx. 560 g was successfully tested at two locations in Germany. Comparative measurements with three identical aerosol samplers on different drones (Matrice 200 and Matrice 300 models, both DJI) showed that the drones themselves and slight differences of the mounting position on the drones had no significant influence on the results. During a measurement campaign (BISTUM23), a series of filters were sampled in parallel using two drones and a ground-based framework, supported by another measurement
UAV that measured the gas phase and meteorological conditions throughout the day. The aim was to perform vertical concentration measurements to investigate the variations in aerosol composition during the course of a day. For this purpose, aerosol samples were collected simultaneously at ground level, 120 m above ground and 500 m above ground.

The primary aim of the measurements shown above is a proof-of-concept, as the amount of data alone is of course not sufficient to make general statements about vertical concentration profiles of organic aerosol components. Nevertheless, some initial
conclusions can be drawn about the measured analytes. The biogenic SOA markers pinic acid, terebic acid, and terpenylic acid show increasing concentrations from the morning hours to the afternoon hours. This result is consistent with the observed increase in ozone concentrations during the day. Both the rising temperatures during the day and thus the increasing release of precursor VOCs during the day and the increasing importance of oxidation reactions can explain this concentration trend (Vettikkat et al., 2023; Niinemets et al., 2004). Interestingly, the vertical concentration measurement showed that the maximum
concentration of these compounds was often observed at a height of 120 m, an observation that may be attributed to different footprint regions, dry deposition and chemical ageing (Spielmann et al., 2017; Bamberger et al., 2011). The highest concentrations of anthropogenic markers were observed in the morning hours. The wind data (Fig. S4 and Fig. S5) and HYSPLIT back trajectories (Fig. S6) indicate that this phenomenon may be due to the main road during rush hour. In general, the concentration of anthropogenic marker compounds is lower than that of biogenic compounds, which can be explained by
the remote location of the sampling site, where biogenic processes can have a greater influence than anthropogenic activities. The results of the non-targeted analysis of the filter samples are consistent with the trends identified in the targeted analysis and show an increase of oxidized compounds throughout the day and with increasing altitude. Consistent with the targeted approach, compounds associated with automobile exhaust and biomass combustion products are particularly present in the morning samples. In summary, this study highlights the use of drones as an innovative platform for the sampling and chemical
characterization of organic aerosols using UHPLC-MS. The developed sampling unit collects sufficient aerosol mass within the drones' flight time, enabling both targeted and non-targeted analysis of primary and secondary organic aerosols. This approach facilitates cost-effective, height-selective sampling, allowing the measurement of vertical concentration profiles and access to otherwise challenging or inaccessible locations for aerosol sampling.



Author contribution

CB and TH developed the filter holder and planed the measurements; CB performed the analytical measurement of the OA samples, analyzed the data and wrote the manuscript draft; BG, NK and TH performed the drone flights; LM performed the flights of FLab and analyzed the ozone and wind data; TH, LM, BG and NK reviewed and edited the paper.

Competing interests

The authors declare that they have no conflict of interest.

Data availability

The measured concentration of the targets can be found in the supplement. The non-target data is available upon request from the corresponding author, Thorsten Hoffmann (t.hoffmann@uni-mainz.de).


Acknowledgements: This work was funded by the Deutsche Forschungsgemeinschaft (DFG, German Research Foundation) – TRR 301 – Project-ID 428312742 and HO 1748/24-1 (Project-ID 541033130).

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
