# Peer review of "Development and use of a lightweight sampling system for heightselective UAV-based measurements of organic aerosol particles"

_EGUsphere, 2024_

## Author Response (AR1)

Dear Editor,

While conducting additional tests in connection with questions raised by Reviewer 3, we used a measurement system that was available to us in the meantime and found a higher airflow than was stated in the previously submitted version of the manuscript. We have incorporated the resulting changes to the flow specifications in the version resubmitted in response to the reviewers' comments. These numerical changes do not affect the overall statement of our manuscript but merely result in an improved detection limit due to the higher flow rate. Furthermore, the ratios between the measured concentrations remain completely unaffected. The increased flow rate only affects the value of the absolute concentrations, which is not relevant to the other statements made in this manuscript. Nevertheless, the work involved in recalibration was more extensive, and we have therefore decided to add another co-author to the list of authors. All co-authors have agreed to this procedure.

We have therefore adjusted our findings accordingly:

Authors list: We added David Wasserzier as co-author.

Line 14: "The sampler consists of a three-dimensionally printed filter holder connected to a lightweight high-performance pump that can generate a flow rate of up to 103 SLPM for up to 30 minutes."

Line 78: "To check the stability of the flow rate through the sampling unit, a filter holder (equipped with PallflexTM EmfabTM filters TX40HI20WW, 70 mm) was connected to a flow meter (model 4043, TSI GmbH, USA) and the flow rate was recorded at 30-second intervals over a total period of 30 minutes."

Line 123: "Figure 2 shows the airflow through the filter holder as a function of time. During the measurement, the recorded airflow decreases from 103.4 SLPM to 92.1 SLPM at an ambient air pressure of 980 hPa."

Line 127: "A linear fit (y = mx + b) was performed, with  $m = (-0.41 \pm 0.01)$  SLPM min-1 and  $b = (102.9 \pm 0.2)$  SLPM as fit parameters."

Figure 2 (a) has been changed and now uses the recalibrated flow.

Line 139: "These constraints result in an operational time of 20 to 30 minutes and a flow rate of approximately 100 SLPM."

In Figure 4 we have adjusted the absolute concentrations.

In Figure 5 the absolute concentrations were adjusted.

In Figure 6 the absolute concentrations were adjusted.

In the supplement the absolute concentrations given in S2, S3 and S4 were adapted.

**RC1: Anonymous Referee #1**

The authors describe a lightweight UAV system for measuring organic aerosol particles and present first vertical measurements from a rural area in southern Germany. The paper is well written and fits within the scope of AMT. However, there are some problems that need to be corrected before the manuscript can be published.

We thank the Anonymous Referee #1 for the supportive review and the constructive comments/suggestions that helped to improve our manuscript. We have carefully revised the manuscript accordingly. Below you will find our point-by-point responses. Reviewer comments and suggestions are written in **black**, **responses in blue**. Changes in the manuscript are marked with "".

**Major comments**

**Introduction**

Why does the paper only mention vertical profile measurements? UAVs are particularly suited for horizontal measurements, which supply the benefit of scanning areas and understanding emission sources (see literature).

Thank you for this comment. We have now added horizontal measurements to the introduction.

Line 54: "In addition to acquiring vertical profiles, the utilization of UAVs enables measurements to be taken in difficult-to-access areas, such as volcanic plumes, or over larger areas at the same height, helping to characterize emission sources (Karbach et al., 2022; Kuantama et al., 2019)."

**Experimental procedures**

Have you studied the turbulence of the rotor? Most groups working on aerosol sampling with UAVs use smog experiments to make turbulence visible. The position of your setup is given in the appendix. However, the pictures don't show the dimensions. Please add these dimensions to the pictures or provide a sketch with the dimensions. This is especially important for the distance between the instrument and the rotors. Since the results are very impressive and the effect of turbulence can obviously be neglected, the question is why. After all, the body of the UAV acts as a kind of shield and allows a shielding flow around the UAV? A colored smog test would show this.

Thank you for this comment. We have not conducted any studies for the turbulence of the rotors, as the experiments show no influence of those on the measured concentrations. A study by Crazzolara et al. (2019) (https://doi.org/10.5194/amt-12-1581-2019) for a larger UAV shows that because of the downwash, air mass from up to about 2 m above the UAV can also be collected. This has no significant influence on the trends shown in this study, as the difference between the measurement points is at least 120 m. In addition, we do not expect large concentration differences within a few meters, so mixing in this area should not significantly affect the measured results. A colored smoke test would provide a more comprehensive picture, but we do not think it is necessary in the context of this study. We have included this in the text and added the dimensions to the pictures.

Line 176: "A study by Crazzolara et al. (2019) investigated the airflow around a larger UAV than the one used in this study, using a colored smoke test. Their findings indicated that the air mass within a radius of up to two meters above the UAV can be affected by the downwash from the UAV. This phenomenon is therefore presumed to exert only a negligible influence on the measured concentrations in a mixed atmospheric boundary layer, as is the case with our results."

Figure 1 shows a schematic of the filter holder. Again, the dimensions are missing. How long is the filter holder? What is the length and diameter of the pump? What is the diameter of the pump inlet? What is the distance from the filter to the pump inlet?

You are right. The dimensions are a relevant factor and should be included in the manuscript. We included them in the schematic of the filter holder (figure 1). The filter holder is approximately 6 cm. The pump measures 2 cm in length, 6.5 cm in diameter, and the inlet diameter is 2 cm. The distance between the filter and the pump inlet is 2.5 cm. The inlet diameter of the pump is now included in the text.

Line 69: "For aerosol sampling, a home-made 3D-printed filter holder (polylactic acid) was connected via a plug to an electric fan motor with an impeller (CDS-R540-QA012; DC 7.2 V; 70 W; inlet diameter 20 mm, SIP Cinderson Motor CO., LTD), which enables high gas flow rates."

Have you determined the deposition characteristics of your filter holder? What are the diameters of the particles (size range) being filtered? Have you determined the total number of particles you are filtering per volume? What are the particle concentrations in number and mass?

We used the PallflexTM EmfabTM filters TX40HI20WW because they have an attested aerosol retention of 99.95 % for following ASTM D 2986-95A 0.3  $\mu$ m (DOP) at 32 L/min/100 cm2 of filter media. Therefore, we did not determine the deposition characteristics of the sampler system and assumed a nearly complete deposition of all present particles.

Other parameters that are missing are the down wash distribution, the rotor speed, and the weight of the sampling unit.

We assume that the downwash distribution does not have a significant effect on our measured concentrations, as mentioned in a previous comment. The rotor speed depends on different parameters like wind speed and air temperature, so it is not known. The total weight of the sampling unit is given in line 75 and in line 295, (sampler plus battery: 560 g).

Can you open and close the inlet to your filter assembly or can you turn the pump on and off during the flight to ensure that you are sampling only from the appropriate altitude?

Thank you for this comment. So far we cannot open or close the inlet or turn the pump on during flight. This is planned for future development of the filter system. Currently, we activate the pump on the ground before proceeding directly to the measuring point. This process takes approximately one minute for the 120 m sample and two minutes for the 500 m sample. This introduces some error, but we still collect most of the time at the appropriate altitude, so we don't consider the influence to be significant. We have specified this in the text.

Line 94: "The pump of the collector was activated on the ground, and the UAV was subsequently flown directly to the designated collection height. This process typically required between one and two minutes to complete."

Electronic circuits and the schematic structure of the measuring unit are also missing. If software was used for the control system, this must also be specified in the appendix.

Thank you for this comment. Since the pump's motor was only connected to a battery by a plug, we don't think it's necessary to provide an electronic circuit. This is now more clearly stated in the

text. Figure 1 shows the schematic structure of the system. It was no software used to control the system.

Line 69: "For aerosol sampling, a home-made 3D-printed filter holder (polylactic acid) was connected to an electric fan motor with an impeller (CDS-R540-QA012; DC 7.2 V; 70 W; inlet diameter 20 mm, SIP Cinderson Motor CO., LTD), which enables high gas flow rates. The electric fan motor is directly plugged into the outlet ports of the battery."

**Results and Conclusions**

The manuscript presents an impressive proof of concept. The results are robust and in line with expectations for the measurement situation. However, more information on the setup is needed to evaluate the errors.

Thank you for your supportive comment. We changed the text in accordance with the earlier comments. It now hopefully contains enough details to better evaluate possible errors.

**Minor comment**

"UAV" is the common term. You could use "UAV" instead of "drone".

Thank you for your comment. We have reviewed the manuscript and changed this accordingly. (Lines: 2, 13, 17, 53, 54, 56, 62, 75, 88, 89, 93, 94, 96, 98, 138, 156, 161, 162, 163, 174, 175, 176, 177, 178, 183, 188, 291, 295, 297, 297, 298, 299, 300, 319, 321 and 327. As well as in the supplement in lines: 3, 10, 12, 12, 13, 14, 15, 23, 57, 75, 80 and 84)

**RC2: Anonymous Referee #2**

In the manuscript, authors are presenting interesting results for atmospheric organic aerosols sampled with lightweight filtration system in the aerial drone. The manuscript is mostly well done and written, although some minor inaccuracies exist and need to be fixed.

We thank the Anonymous Referee #2 for the supportive review and the constructive comments/suggestions that helped to improve our manuscript. We have carefully revised the manuscript accordingly. Below you will find our point-by-point responses. Reviewer comments and suggestions are written in **black**, **responses in blue**. Changes in the manuscript are marked with "".

**Page 3, line 65: What was the material used for printing the filter holder?**

Thank you for this comment. We have added this information.

Line 69: "For aerosol sampling, a home-made 3D-printed filter holder (polylactic acid) was connected via a plug to an electric fan motor with an impeller (CDS-R540-QA012; DC 7.2 V; 70 W; inlet diameter 20 mm, SIP Cinderson Motor CO., LTD), which enables high gas flow rates."

Page 3, line 83: Generally the best sampling point is under the body of the drone/sampling unit (less affected by the flow caused by propellers)?

The best sampling location would be about 1.5 m in front of the drone as, at this location, the air is not affected by the downwash of the propellers anymore. However, mounting the sampler at this location proves to be extremely challenging. Furthermore, we found no difference for the different types of UAVs, different mounting locations on the UAV and also no difference when comparing samples with and without a drone as shown in Figure 4. We therefore concluded that the downwash from the propellers has only a negligible influence on the sampled amount and type of air mass.

Page 4, lines 96-99: Total volume of extraction solutions will be 3mL+1.5mL+1.5mL. How this (6 mL together) can be transferred to 1.5mL vial? Rewrite to clarify what was done and how. Also give the volume to what the supernatant was concentrated under N2

We have adapted the section to hopefully make it clearer.

Line 106: "The supernatant was successively transferred to a 1.5 mL HPLC vial and concentrated to approximately 50  $\mu$ L at 30 °C under a gentle N2 stream. The residue was then filtered with a PTFE filter (pore size: 0.20  $\mu$ m; Altmann Analytik)."

Page 4, lines 105-110: Generally, both A and B eluent components should contain 0.04% formic acid. Then its amount (and eluent pH) would stay constant. Now the pH will change during the concentration gradient. Also you should report what was the pH. For acidic compounds with negative ESI, basic conditions are normally used for efficient ionization and good sensitivity. Now the conditions used are more suitable for positive ESI. You could have tried to modify the eluent and pH used for example adding ammonium acetate or ammonium fluoride instead of formic acid. If the chromatography is not good with these, then this modification could be done post column with ESI nebulizing liquid.

Thank you for your feedback. It is correct that the additives in the eluent are better for ESI positive measurements. These additives are essential for obtaining a good peak shape and also for obtaining a good separation of the organic acids. It's also true that the pH value changes during the measurement. However, this is not a problem in our study because we used an external

calibration to determine the concentrations. This ensures that the pH value is the same for both standards and samples and therefore the actual pH value does not play a role. We do not determine concentrations for the non-target evaluation, so the pH value's influence on ionization efficiency is negligible. A post column flow could improve the ionization efficiency of the compounds, which would increase the detection limit. As the manuscript shows, this is not necessary for the compounds investigated, as they show a high intensity with a good signal-to-noise ratio even without post column flow. However, it would be interesting to use post-column flow in the future to see if it has a positive effect on signal intensity.

Page 4, lines 116-117: The voltage regulator would be good to include in the future (and maybe mention this in the conclusions).

This has been added to the text.

Line 321: "A voltage regulator could be integrated in the future to ensure a constant flow through the aerosol sampler."

Figure 7, figure legend: 10:35 is as far from the noon as is 13:35. You could talk about late morning (10:35am), early afternoon (13.35pm) and late afternoon (4:30pm)? Also for some cases you have 24h system (13:35 pm) and some 12 h system (4:30 pm), so be consistent. Fix these throughout the whole manuscript (at least in Figure 6 there is 13:35 pm).

Thank you for this comment. We have carefully reviewed the manuscript and figures and corrected this issue.

Line 287: "Figure 7: Van Krevelen diagrams in the late morning (left), in the early afternoon (middle) and in the late afternoon (right). The different substance classes are represented by different colors (CHO blue; CHON green; CHOS orange; CHONS pink) The size of the dots correlate with the measured peak area of the compounds. The Van Krevelen diagrams are divided into 5 areas depending on the MCR, which are separated by dashed lines (Zhang et al., 2021)."

**RC3: Anonymous Referee #3**

Borchers et al. provide a layout for an aerosol sampling system aboard drones. The manuscript is well written and the results are very nice, especially the untargeted analysis. I have a couple of major comments, but otherwise think that this is a nice paper that sets the ground for future work.

We thank the Anonymous Referee #3 for the supportive review and the constructive comments/suggestions that helped to improve our manuscript. We have carefully revised the manuscript accordingly. Below you will find our point-by-point responses. Reviewer comments and suggestions are written in **black**, **responses in blue**. Changes in the manuscript are marked with "".

**General comments:**

**Title:** My major question is why the manuscript focuses so heavily on organic aerosol? Surely this sampling system is appropriate for submicron aerosol of any composition? One could analyse ammonium, nitrate, and sulfate on these filters via ion chromatography. I think a more appropriate title would be Development and use of a lightweight sampling system for height selective drone-based measurements of organic aerosol particles"

Thank you for this comment. It may be possible to collect inorganic components with this system. The collection, extraction, and evaluation method shown here has not been tested for this particular application, so it is outside the scope of this paper. In the interest of being clear about the paper's focus, which is on organic aerosols, we have decided to include this in the title.

**Flowrates:** Figure 2 is very worrying—the flow drops hugely across a 30 minute period. Was this data taken in the lab? Were any flowrate measurements taken when the UAV was in the air? Was any flowrate correction made considering that atmospheric pressure is substantially lower at 500 m? I don't think any of the results are trustworthy if the flowrate is this unsteady and unmeasured. There's no reason to believe any of the quantifications as they totally depend on the sampled volume. This also applies to the error bars in Figure 5. You've decided there is a 2% error, but that would only apply if there was uncertainty but no trend downwards. There's no reason to believe that the data from Figure 2 wasn't taken at a sample rate of 10 L min-1.

**Thank you for this comment.**

You are correct that the flow drops over the 30 min interval. The drop is about 10%. This is to be reduced in the future in the further development of the filter holder shown here by installing a voltage regulator. Since this drop has been taken into account when determining the concentrations, we see no problem with the drop. The flow was determined in the laboratory and no measurements were taken during the flight. We have now made measurements at different air pressures to determine the dependence of the flow on air pressure and to be able to correct the measurements accordingly (Figure 2 (b) and supplement).

Supplement line 38: "The collected air volume ( $V_{\rm air}$ ) is then determined by integrating the linear equation of the fit for the flow (Q) (Equation S2) through the filter holder, which leads to Equation S3. In the following equations t is the sampling time.

$$Q = (-0.41 \pm 0.01) \text{ SLPM min}^{-1} \cdot t + (102.8 \pm 0.2) \text{ SLPM}$$
 (S2)

$$V_{\text{air}} = \frac{(-0.027 \pm 0.002)}{2} \text{ SLPM min}^{-1} \cdot t^2 + (18.97 \pm 0.04) \text{SLPM} \cdot t$$
 (S3)

Since the flow through the collector is pressure dependent, it must be corrected for the different pressures at different heights (1.5 m: 920 hPa; 120 m: 907 hPa; 500 m: 870 hPa; measured by FLab). The linear fit for the dependence of flow on pressure (Q(p)) is used for the correction (Equation S4).

$$Q(P) = (0.20 \pm 0.01) \text{ SLPM hPa}^{-1} \cdot p + (-93 \pm 12)$$
 (S4)

The collected air volume  $V_{\rm air}(p)$  is subsequently corrected according to Equation S5. The flow at 980 hPa is utilized as a reference because the time dependence was determined at this air pressure.

$$V_{\text{air}}(p) = V_{\text{air}} \cdot \frac{Q(p)}{Q(980 \text{ hPa})} \tag{S5}$$

The concentration of the compound of interest can then be calculated as shown in Equation S6.

$$c(\text{compound}) = \frac{m(\text{compound})}{V_{\text{air}}(p)}$$
 (S6)"

Line 126: "A linear fit (y = mx + b) was performed, with  $m = (-0.41 \pm 0.01)$  SLPM min-1 and  $b = (102.9 \pm 0.2)$  SLPM as fit parameters. In an additional experiment a second linear fit for the relationship between the flow rate and the ambient pressure was performed this yielded the linear fit ( $\dot{V}(p) = m_p \cdot p + b_p$ ) with the parameters  $m_p = (0.20 \pm 0.01)$  SLPM hPa-1 and  $b_p = (93 \pm 12)$  SLPM."

Line 133: "Figure 2: (a) shows the flow through the filter holder as a function of time (at 980 hPa) (blue dots), the dashed line represents a linear fit of the data. The errors correspond to 2% of the measured value, which represents the measurement uncertainty of the flow meter (b) shows the flow through the filter holder as a function of pressure (green dots), the dashed line represents a linear fit of the data. The errors correspond to 2% of the measured value, which represents the measurement uncertainty of the flow meter."

**Specific comments**

Line 8: Do you mean natural and anthropogenic sources?

Yes, we meant natural and anthropogenic sources and have changed it in the text.

Line 9: "Organic aerosols (OA) are introduced into the atmosphere from a variety of natural and anthropogenic sources."

Line 73: Will being airbourne have any effect on the stability of the flows?

Thank you for your comment. We have investigated the influence of the different drones or when no drone is used. The results in Figure 4 show that there is no significant influence of the drone on the flow.

Figure 1: this figure is missing some details about dimensions etc.

You are right. We included the dimensions in Figure 1.

Line 91: I know it's not a major part of the manuscript, but if you briefly described the FLab system (one or two sentences) it would be useful so we do not have to go to Moormann et al. (2024).

We added a clarification about the benefits of simultaneous FLab measurements to the paper:

Line 98: "A third measurement UAV, FLab (Moormann et al., 2025), was used to simultaneously quantify gas tracers and meteorological data in hourly vertical profiles over the course of a day at the same measurement location. In particular, height-resolved monitoring the  $O_3$  mixing ratio and

wind conditions within 500 m range above ground show oxidative potential of air and supports attribution of air mass origin. "

Line 119: Why is the linear fit used here? Surely you can just integrate your points and get a more accurate value.

Thank you for this comment. We used the linear fit method to account for any measurements that did not fall exactly on one of the measured points for the flow determination. In addition, its influence is minimal and compensates for slight flow fluctuations during the measurement.

Line 176: UTC, not UCT.

We have corrected this.

Line 91: "Figure 5 shows three height profiles at 10:35 am, 1:35 pm and 4:30 pm local time (UTC+2) for concentrations of the three biogenic marker components pinic acid (black square), terpenylic acid (blue triangle) and terebic acid (green diamond) in 1.5 m, 120 m and 500 m."